# Advancements in Chemical and Biosensors for Point-of-Care Detection of Acrylamide

**DOI:** 10.3390/s24113501

**Published:** 2024-05-29

**Authors:** Mingna Xie, Xiao Lv, Ke Wang, Yong Zhou, Xiaogang Lin

**Affiliations:** Key Laboratory of Optoelectronic Technology and Systems of Ministry of Education of China, Chongqing University, Chongqing 400044, China; mingnaxie@stu.cqu.edu.cn (M.X.); 202308021083t@stu.cqu.edu.cn (X.L.); kewang@stu.cqu.edu.cn (K.W.)

**Keywords:** sensors, acrylamide, POC, optical, electrochemical, food safety

## Abstract

Acrylamide (AA), an odorless and colorless organic small-molecule compound found generally in thermally processed foods, possesses potential carcinogenic, neurotoxic, reproductive, and developmental toxicity. Compared with conventional methods for AA detection, bio/chemical sensors have attracted much interest in recent years owing to their reliability, sensitivity, selectivity, convenience, and low cost. This paper provides a comprehensive review of bio/chemical sensors utilized for the detection of AA over the past decade. Specifically, the content is concluded and systematically organized from the perspective of the sensing mechanism, state of selectivity, linear range, detection limits, and robustness. Subsequently, an analysis of the strengths and limitations of diverse analytical technologies ensues, contributing to a thorough discussion about the potential developments in point-of-care (POC) for AA detection in thermally processed foods at the conclusion of this review.

## 1. Introduction

Thermal processing stands as a ubiquitous method for preserving and packaging food and inevitably produces undesired chemical contaminants in the meantime [1]. Among these undesirable byproducts of food processing is acrylamide (AA), which was classified as a ‘probable carcinogen’ by the International Cancer Agency in 1994. The monomeric form of AA is a toxic compound with potential carcinogenicity, neurotoxicity, genetic toxicity, as well as reproductive and developmental toxicity with high permeability [2]. In 2002, it was demonstrated that plant-based foods high in carbohydrates and low in protein are prone to yield abundant AA during high-temperature (>120 °C) processing [3,4]. The result has attracted wide interest in recent years owing to its toxicity and wide occurrence in most thermally processed carbohydrate-rich foods. There are two mechanisms to form AA in such thermal processing (Figure 1): the Strecker and the acrolein pathways [5]. On the one hand, the Strecker pathway is associated with the Maillard reaction, a non-enzymatic browning process that transpires between amino acids and reducing sugars. In this pathway, amino acids undergo decarboxylation and deamination processes, resulting in the formation of Strecker aldehydes, which eventually contribute to the production of AA [6,7]. On the other hand, the acrolein pathway is characterized by thermolysis, where the decarboxylation of organic acids leads to the generation of AA [8]. Consequently, AA can also be generated in fat-rich foods such as ripe black table olives through this pathway [9,10].

According to the investigation and report of Commission Regulation (EU) 2017/2158, which set benchmark levels of AA in food, the average AA content is in the range of 40–4000 ppb [11]. Commission Regulation (EU) 2019/1888 proposed that foods used to monitor the presence of AA (potato products, bakery products, cereal products as well as dried fruits, marinated olives, etc.) are a crucial part of human food [12]. According to the FAO and WHO, the permitted daily intake of AA in food is 0.3–0.8 ppb [3]. Monitoring and controlling AA levels in food products becomes particularly crucial due to the potential and accumulated health implications of prolonged and repeated exposure to low concentrations of this compound [13].

Conventional methods for AA detection, including standard gas chromatography–mass spectrometry (GC-MS) and liquid chromatography–mass spectrometry (LC-MS), high-performance liquid chromatography (HPLC), capillary electrophoresis, as well as enzyme-linked immunosorbent assay (ELISA), have proven reliable due to their high accuracy, sensitivity, selectivity, and robustness. However, these methods, with drawbacks such as the need for expensive and sophisticated instrumentation, strict environmental conditions, skilled personnel for operation, complex sample preparation, and even being time-consuming [14], hinder the establishment of on-site and point-of-care (POC) AA detection during food processing. As a result, there is a growing demand for rapid analysis, portability, ease of use, robustness, minimal sample requirements, real-time data analysis, affordability, as well as disposable or reusable options in this aspect. Technologies such as immunoassays, sensing techniques [15,16], and microfluidic chips are emerging as promising alternatives.

This review delves into the bio/chemical sensors for the detection of AA in the past decade, with emphasis on the detection materials, detection methods, linear range, limits of detection (LoD), and response time. These bio/chemical sensors employ optical, electrochemical, hybrid, as well as piezoelectric transduction systems. The review evaluates these sensors based on their sensitivity, selectivity, stability, response time, and robustness, highlighting both their strengths and limitations, and forecasting the future expansion and development potential of such sensors within the food industry.

## 2. Principles of Bio/Chemical Sensors for AA Detection

Sensors are devices, systems, or modules designed to detect and quantify target molecules of interest. They typically incorporate specialized receptors, which can be physically close to or linked with a transduction system [17]. Additionally, the receptor converts measurable energy generated during chemical reactions into data, while the transformer further converts this data into useful electrical signals [18].

Due to the electron deficiency in AA’s vinyl group, AA is vulnerable to nucleophiles. Therefore, AA is recognized as a Michael receptor, capable of forming adducts with significant functional groups including thiol groups (-SH), hydroxyl groups (-OH), and amino groups (-NH_2_) found in DNA or other biomolecules [19]. On the other hand, AA is also capable of coordinating with less acidic metal ions to form complexes through the carbonyl oxygen atom, the nitrogen atom, or the olefin in $\eta^{2}$ mode [20]. Table 1 outlines the typical receptors and ligands that bind with AA. Utilizing the aforementioned detection mechanisms, these sensing technologies have been developed in various food substrates (Figure 2).

### 2.1. Optical Sensor for AA Detection

Optical sensors leverage optical field characteristics to translate binding events into detectable and measurable signals when targets bind to the bio/chemical recognition element. Based on optical characteristics, optical sensors for AA detection primarily fall into the following categories (Table 2).

#### 2.1.1. Colorimetric Sensing

Colorimetric measurements usually involve tracking alterations in absorbance resulting from variations in the concentration of the analyte at a particular wavelength [21]. Colorimetric transduction signals primarily utilize nanoparticles (NPs), which, due to their size-dependent optical properties, emit light at specific wavelengths. A colorimetric technique utilizing the aggregation or redispersion of gold nanoparticles (AuNPs) has been utilized for DNA detection. This method is valued for its simplicity, affordability, high sensitivity, swift response, naked-eye visible signal interpretation, and lack of dependence on costly or intricate instruments [22]. Generally, factors that trigger a distance change between AuNPs are the key points to designing a novel colorimetric method. Presently, two primary strategies are employed: biological recognition, such as DNA hybridization [23,24,25] and immune recognition [26,27], alongside physical interactions [28,29,30,31].

For compounds lacking distinctive properties like AA, designing a straightforward colorimetric detection proves challenging. Some specific recognition modes have been reported in this field. For instance, Hu et al. proposed an AuNPs colorimetric method based on a nucleophile-initiated thiol-ene Michael addition reaction between glutathione (GSH) and AA and offered visual detection of AA. The AuNPs were GSH-modified, inducing a color shift from red to purple due to the formation of an Au-S bond between the Au surface and the -SH (in GSH). However, in the presence of AA, a Michael addition reaction competitively occurred between the -SH (in GSH) and the C=C double bond of AA, resulting in the formation of the GSH-AA adduct. This reaction caused the dispersion of AuNPs via a non-cross-linking mechanism, leading to a color change from purple or dark blue to red. The total detection time was 2.5 h, with an LoD of 28.6 nmol L^−1^ [32]. Shi et al. also put forward a colorimetric assay based on AA-mediated polymerization, causing a distance increase between AuNPs. In this method, AuNPs were modified with thiolated propylene amide poly(ethylene glycol) (AAPEG-SH) incorporating the AA function [33]. Nevertheless, the drawbacks of the aforementioned methods lie in their multistep preparation and the complexity of AA recognition. Therefore, Hoang et al. first made use of combining colloidal silver nanoparticles synthesized by an electrochemical method (e-AgNPs) with thiourea (TU) to serve as colorimetric sensors for the direct recognition of AA. While the method demonstrated the directed recognition of AA with an LoD of 0.024 μM and a wide linear range (0.1–1000 μM, R^2^ = 0.99), the presence of similar compounds, especially methylamide, significantly interfered with the accurate determination of AA, contributing to approximately 10% of the signal [34].

Aptamers, unique single-stranded oligomers of RNA or DNA, are selected using the systematic evolution of ligands by exponential enrichment (SELEX) technique. Among various biorecognition elements, aptamers stand out due to their small size, ease of production and modification, cost-effectiveness, non-toxicity, reusability, high thermal and chemical stability, and excellent specificity in binding to diverse targets. These attributes make them widely employed in biosensors to enhance performance. The fusion of nanotechnology with aptamer science enables the development of automated and miniaturized aptasensors, facilitating the on-site monitoring of ultratrace levels of targets. Khoshbin et al. first proposed a portable colorimetric aptasensor for the label-free detection of low levels of AA. As shown in Figure 3A, the immobilized STP strand prevented AuNPs from salt-induced aggregation, maintaining a red solution at the tip. In the absence of AA, the locker strand was bound to the complementary fragment at the end of the specific aptamer. However, in the presence of AA, the locker strand was released, and it hybridized with the STP strand to form a triple-helix molecular switch (THMS) structure, leading to an evident color change from red to blue. Subsequently, a smartphone imaging readout-based strategy was employed to quantify AA through the RGB values outputs (Figure 3B) with an LoD of 0.038 nmol^−1^, and recoveries range from 92 to 102%. In addition to being portable and sensitive, the sensor’s specificity was confirmed by analyzing its response in the presence of interfering compounds such as glycine, aspartic acid, ascorbic acid, acetate, acrylic acid, caffeine, and beta-alanine. In Figure 3C, it was observed that the B value of AA was markedly higher compared to the interfering compounds tested. The designed aptasensor also possessed cost-effectiveness and rapid detection capabilities, with minimal solution consumption [35].

#### 2.1.2. Fluorescence Sensing

Fluorescence spectroscopy, being a straightforward and non-destructive sensing technique, has significantly influenced food quality control due to its simple online monitoring mode. Utilizing chemical/biosensors, fluorescence spectroscopy can offer robust signal transduction for the target analyte through simple operation, boasting high selectivity and sensitivity. These attributes make it an ideal choice for AA detection [36]. The mechanisms of fluorescence quenching include fluorescence resonance energy transfer (FRET), static quenching, dynamic quenching, inner filter effect (IFE), and photoinduced electron transfer (PET) [37].

Earlier works on fluorescence sensors employed some of the simplest chemical reactions to generate fluorescent signals. For instance, Liu et al. developed a fluorescence method for detecting AA utilizing the Hofmann reaction. During the determination process, AA undergoes the Hofmann reaction, producing vinyl amine. Upon the reaction of vinyl amine with fluorescamine, pyrrolinone is generated, leading to a significant fluorescence emission at 480 nm [38]. However, a limitation of this method is the requirement for high temperatures during its fabrication procedure.

Many recent fluorescence sensors rely on the FRET, a phenomenon rooted in the non-radiative energy transfer from an excited donor to an acceptor via an electrostatic dipole–dipole interaction. In addition to fluorescent dyes, conventional chromophores such as quantum dots (QDs) can undergo self-aggregation mediated by non-covalent bonds such as hydrogen bonding interactions, π-π interactions, electrostatic forces, and van der Waals forces. These non-covalent interactions typically render the aggregation process reversible. By triggering the disaggregation of fluorescent QDs through analyte interaction, Hu et al. developed a switchable fluorescent sensor for AA detection based on AA polymerization-induced distance increase between QDs. In this study, the C=C double bonds of N-acryloxysuccinimide (NAS)-modified QDs underwent polymerization catalyzed by a photoinitiator upon UV irradiation, leading to a decreased inter-QD distance and subsequent fluorescence intensity reduction. However, with AA involvement in the polymerization, the inter-QD distance expanded, yielding a heightened fluorescence intensity and achieving an LoD of 3.5 × $10^{-5}$ g·L^−1^. In the selectivity experiment, they found that the addition of acids other than 6-aminocaproic acid (such as succinic acid, acetic acid, acrylic acid, propionic acid, and N-butyric acid) caused quenching due to their influence on the pH value of the environment around QDs, which could result in false negative errors in food sample determination. Additionally, the interference of L-asparagine, the main precursor of AA in thermally processed foods, requires the use of L-asparaginase to avoid false positive errors in the detection of food samples. Nevertheless, the sensitivity and selectivity of this method fall short compared to traditional LC-MS/MS analysis and are unsuitable for determining low AA concentrations in food [39]. Carbon quantum dots (CQDs) demonstrate a promising performance as FRET donors or acceptors due to their environmentally friendly nature, mild toxicity, cost-effectiveness, distinctive wide absorption, and excitation-dependent emission properties. Wei et al. pioneered the utilization of CQDs for AA detection in food, presenting a straightforward AA quantification. They achieved this by leveraging the fluorescence enhancement triggered by the expansion in distance between functionalized CQDs modified by NAS-modified CQDs (NAS-CQDs) during AA polymerization, using an optimized concentration of AA as the initiator. This chemical sensor exhibited no interference and possessed an LoD of 2.6 × 10^−7^ M in the double-distilled water system and of as low as 8.1 × 10^−7^ M after applying the QuECHERS (Quick, Easy, Cheap, Effective, Rugged, and Safe) method for white bread crust [40]. Colloidal gold, known as AuNPs, has become widely utilized in biosensing applications for its efficacy as a fluorescence quenching agent. This stems from its attributes such as high chemical stability, ease of synthesis, high selectivity, and high absorption coefficient [41]. When the emission spectrum of CQDs aligns with the absorption spectrum of AuNPs in the blue region, efficient FRET-based quenching occurs. Acknowledging the surface-functionalization potential of both CQDs and AuNPs, Pattnayak et al. constructed a FRET-based sensing platform utilizing -SH-functionalized CQDs as donors and citrate-stabilized AuNPs as acceptors, with an LoD of 0.1 pM. In Figure 4A, molecular recognition between the FRET sensor and AA has been achieved through the thiol-ene Michael addition reaction. Besides, the color of the CQD-Au nanoprobe solution under long-wavelength UV could be converted to RGB values (Figure 4B), which were used for the quantitative analysis of the AA using a smartphone. Figure 4C demonstrates a linear relationship between AA concentration over a linear range of 10–80 nM and a correlation efficiency of 0.9859 for App-based AA detection. The functional nanoprobe was subjected to selectivity and interference tests using various structurally similar molecules (sucrose, Listine, L-alanine, D-glucose, acetic acid, acrylic acid, and methylacrylamide) and suspected cations (Na^+^, K^+^, Ca^2+^, Fe^3+^, Cu^2+^, and Ag^2+^). The results indicated that the probe exhibited superior selectivity and more accurate detection of AA. Furthermore, the proposed analytical method showcased a shorter processing time (<10 min) compared to the HPLC method (<60 min) [42].

The inner filter effect (IFE) represents a non-irradiation energy transfer model in spectrofluorometry, arising from the absorption of excitation or emission light of the fluorophores by the absorber. As changes in the absorption intensity of the absorber lead to exponential variations in the fluorescence intensity of the fluorophore, the IFE has proven to be an effective and potent strategy for enhancing the detection sensitivity [43,44]. Luo et al. proposed an IFE-based fluorescent immunoassay by integrating the IFE-based alkaline phosphatase (ALP) sensing system with an ALP-based ELISA platform, and the LoD was 0.16 μg/L. The fluorescence immunoassay was subsequently employed to analyze AA in drinking water, biscuits, and potato chips using the standard addition method. Given the complex matrices of food samples, extracts were diluted 50-fold to mitigate matrix effects. Ultimately, the recoveries for drinking water, biscuits, and potato chips were 82.3% to 93.5%, 81.0% to 105.6%, and 88.7% to 103.2%, respectively. This fluorescence immunoassay provided a potent strategy for AA detection [45].

Despite the high sensitivity, selectivity, simple operation, and rapid detection, most fluorescence analysis methods rely on a single signal response involving fluorescence enhancement or quenching [46], which is susceptible to environmental interference. Ratiometric fluorescence, however, represents a strategy for measuring the ratio of emission intensity at two different wavelengths simultaneously. Porphyrin Metal–Organic Frameworks (MOFs) exhibit the promising absorption of dye-labeled oligonucleotide strands through π-π stacking interactions and subsequent quenching of the dyes via both a photoinduced electron transfer (PET) process and FRET, making them effective quenchers [47]. Consequently, Gan et al. designed a ratiometric fluorescence biosensor based on a 6-FAM labeled aptamer (FAM-ssDNA) and porphyrin MOFs (PCN-224). The FAM-ssDNA adhered to the surface of PCN-224 via π-π stacking, hydrogen bonds, electrostatic interactions, and coordination interactions, leading to fluorescence quenching. Upon hybridization with complementary ssDNA (csDNA), the conformational change in FAM-ssDNA restored its fluorescence. While, in the presence of AA, the conjugation of AA and FAM-ssDNA inhibited the formation of FAM-dsDNA. The PCN-224 biosensor leveraged the specific recognition between AA and the guanine base of its aptamers, resulting in significant selectivity and making it suitable for detecting a complex real sample matrix. Besides its sensitivity and selectivity, the sensor exhibited good reproducibility, with an RSD range of 2.7% to 4.4%. This highlighted its broad application prospects in food and environmental analysis [48].

Cheng et al. recently proposed a ratiometric fluorescence sensor based on copper nanoclusters (CuNCs) for AA detection in food. This study synthesized CuNCs with bimodal emission using bovine serum albumin (BSA) as the ligand and ascorbic acid (ASA) as the reducing agent. These CuNCs displayed optimal excitation at 310 nm and maximum fluorescence emission at 650 nm, along with bimodal emission at both 395 nm and 650 nm. GSH was found to quench the fluorescence emission at 650 nm of CuNCs while enhancing it at 395 nm, confirming a dynamic quenching. In the presence of AA, the formation of an adduct between AA and GSH restored the fluorescence emission at both 650 nm and 395 nm, with the LoD of 1.48 μM. The sensor demonstrated robust detection results even in complex environments with interferences such as methacrylamide, acrylic acid, propionic acid, DL-alanine, glycine, potassium sorbate, L-asparagine, sucrose, glucose, and fructose. This indicates the system’s high selectivity and anti-interference capability. When further used to detect AA in toast, the sensor achieved recoveries ranging from 90.29% to 101.30%, with an RSD between 2.18% and 3.31% [49].

#### 2.1.3. SERS Sensing

Surface-enhanced Raman spectroscopy (SERS) has emerged as a powerful molecular spectroscopic technique with an ultrasensitive detection capability [50,51]. In SERS, a Raman spectrum is obtained based on the Raman shift, where each peak in the spectrum corresponds to the vibration of a specific molecular bond, facilitating the identification of target analytes [52]. Moreover, SERS enables a significant enhancement of the Raman signal, up to $10^{6}$-fold [53]. Hence, Gezer et al. examined a biodegradable zein/gold SERS platform as a potential tool for AA detection for the first time. Their findings revealed that the presence of AA resulted in a characteristic labeled SERS peak at 1447 nm that is absent in the SERS spectrum of the sensor background. While the efficiency of the sensor across various food substrates remains unknown, this proof-of-concept underscores the potential of SERS sensors engineered on biodegradable platforms [54].

SERS enhancement operates via two mechanisms: electromagnetic enhancement (EE) [55] and chemical enhancement (CE) [56]. EE suggests that, under light irradiation, free electrons generate a robust electromagnetic field on the surface of rough metal substrates (Ag, Au), amplifying the molecular-induced dipole moment on the substrate surface. This, in turn, increases the probability of molecular Raman scattering, thus enhancing the Raman signal [57]. Moreover, CE suggests that the molecular electronic structure and polarizability affect the Raman intensity through phenomena such as surface chemical adsorption, surface complex formation, and charge transfer [58,59]. Wang et al. believed that the strong electromagnetic field near metallic nanoparticles plays a major role in the occurrence of SERS compared to CE. So, they reported the fabrication of raspberry-like polydopamine (PDA)/AgNPs composites (PDA@AgNPs), wherein PDA spheres served as multifunctional reaction templates, binding reagents, and reducing agents. The raspberry-like Ag composites with good SERS sensitivity were employed for AA determination in water, demonstrating a low LoD of 0.04 g/L. Even so, the PDA@AgNPs-based SERS method only verified the anti-interference performance in the presence of possible coexisting ions (Fe^3+^, Al^3+^, Mn^2+^, Cu^2+^, Zn^2+^, Na^+^, Cl^−^, ${\mathrm{NO}}_{3}^{-}$ and ${\mathrm{SO}}_{4}^{2-}$), and the resulting SERS spectrum showed no significant effect on the AA detection. Furthermore, the substrates exhibited a storage time of up to two months. However, it is worth noting that the method relies on portable Raman equipment [60].

Nonetheless, two technical challenges hinder the practical application of SERS: achieving high sensitivity and reproducibility. To address the former, considering that local surface plasmon resonance (LSPR) often occurs in the gaps between nanostructures (such as AuNPs and AgNPs) to generate SERS-enhanced “hot spots”, substrates with a high density of hot spots can be designed. However, the unmanageable aggregation of metal NPs leads to the random formation of hot spots, resulting in inhomogeneous SERS enhancement. Additionally, the graphene-based substrate can form large π bonds due to its sp^2^-hybridized carbon atoms, allowing the target compound to be uniformly adsorbed on the surface. This process, known as the graphene-enhanced Raman scattering (GERS) effect, can provide uniform Raman enhancement and improve the stability of the composite substrate. Cheng et al. thus proposed a method for determining AA content in fried food based on SERS with reoxidized graphene oxide/AuNPs composites with an LoD of 2 μg·kg^−1^ and limits of quantification of 5 μg·kg^−1^. To enhance anti-interference and reduce the matrix effect, this study employed dispersive solid-phase extraction (dSPE) to pretreat samples and used matrix-matched calibration standard curves to quantify AA in fried food. The substrates exhibited a long-term stability of approximately 180 days at 4 °C, while the total detection time per sample was only 9.5 min. The proposed method and LC-MS/MS were both used to detect AA in three types of fried food collected from China. The results were highly consistent, indicating that the method possesses the potential to be used for the on-site detection of AA [61].

Now, it is generally believed that SERS results from the combination of EE and CE mechanisms. Wu et al. pioneered a sensitive filter paper-based substrate utilizing strawberry-like SiO_2_/AgNPs (F-SANC) for the first-time detection of AA, achieving an LoD of 0.02 nM. This method revealed a characteristic peak at 1630 cm^−1^. The functional substrate, integrating SiO_2_ nanoparticles (SNPs) to provide a 3D supporting substrate for AgNPs (SANC) and modifying the SANC surface with polyvinylpyrrolidone (F-SANC), generated numerous “hot spots” and specific adsorption surfaces, facilitating SERS signals through both EE and CE. To facilitate the loading of SANC on the filter paper, the F-SANC substrates were constructed via a dipping method, which exhibited good reproducibility (a variation coefficient of 6.2%). Moreover, the SERS intensity of the F-SANC substrate remained at about 87.2% of the original after 6 months of storage in 10 mL polypropylene tubes at 4 °C. Nevertheless, this study only investigated three types of food samples, and the selectivity relied heavily on characteristic SERS bands [13].

Recently, Ye et al. developed a simple, rapid, and convenient SERS method coupled with a substrate and an aggregating agent for AA detection in fried food. They investigated various SERS substrates (AuNPs and AgNPs), AgNPs combined with different aggregating agents (NaCl, KCl, MgCl_2_, Na_2_SO_4_, and MgSO_4_), and optimized proportions of the aggregating agents to enhance the Raman signal (Figure 5A–H). They utilized the characteristic Raman peak of AA at 1449 cm^−1^ to determine the optimal quantities of the analyte, aggregating agent, and AgNPs through an orthogonal experiment. Finally, the SERS analysis of potato chips as real food samples was conducted using an AgNP substrate and 0.5M NaCl as a coagulant. The LoD was 2.5 μg/L, and the recovery rate ranged from 94.67% to 117.50%. After verification by LC-MS/MS, further studies are needed to assess the anti-interference capabilities of this method and its performance in other food samples [62].

### 2.2. Electrochemical (EC) Sensor for AA Detection

Electrochemical (EC) sensors are devices that convert information received during electrochemical reactions between electrodes and analytes present in sample solutions into useful quantitative and qualitative electrical signals. These sensors operate by detecting electrical signals generated during redox reactions between the target analyte and the transduction material on the sensing electrode, thereby providing valuable analytical data [63]. The EC sensors are divided into these major types depending upon different types of electrical signals and principles of workability [64]: potentiometric, conductometric sensor, impedometric, coulometric, electrochemiluminescent, and voltammetric sensor. As can be seen from Table 3, recent advancements in EC sensors for AA detection predominantly utilize voltammetry, so this section will introduce them from the perspective of the probe.

**Table 2 sensors-24-03501-t002:** Bio/chemical sensors based on optical transduction.

Sensor Type	Test Material	Linear Range	LoD	Time	Average Recovery (%)	Food Sample	Year, Refs
Colorimetric	AuNPs modified with GSH	0.1–80 μmol/L	28.6 nmol/L	1 min	-	Potato chips	2016, [32]
	AuNPs modified with PEG	-	0.2 nM	-	98.8–109.1	Potato chips, baked cookies, and non-fried cookies	2018, [33]
	e-AgNPs modified with TU	0.1–1000 μM	0.024 μM	-	82–90	Biscuits	2021, [34]
	THMS structure of DNA strands	0.038 nmol/L	0.05–200 nmol/L	-	92–102	Chips, coffee, and bread	2022, [35]
	MGzyme-csDNA	0.01–100 μM	1.53 nM	50 min	99.00–104.4	Artificial meat, biscuits, and potato chips	2023, [65]
	Fe-PHS nanozyme	0.75–36.00 μM	0.27 μM	1 h	87.72–112.87	Chips, coffee, and bread	2023, [66]
Fluorescent	CSUCNPs modified with csDNA of AA aptamers	0.001–10 μM	1.00 nM	30 min	-	Potato chips	2024, [67]
	FAM-ssDNA and PCN-224	10–0.5 mM	1.9 nM	-	94.7–104.3	Potato chips and biscuits	2022, [48]
	CuNCs	5–300 μM	1.48 μM	5 min	90.29–101.30	Toast	2023, [49]
	ALP-based ELISA platform	0.21~6.48 μg/L	0.16 μg/L	-	81.0–105.6	Drinking water, cookies, and potato chips	2021, [45]
	FAM-csDNA	0.67–16.7 μM	0.16 μM	-	95–110	Potato chips	2022, [36]
	CQD-Au nanoprobe	0–200 nM	0.1 pM	10 min	98.6–102.6	Fried bread sticks and potato chips	2023, [42]
SERS	Biodegradable zein/gold SERS platform	-	-	-	-	-	2016, [54]
	PDA@AgNPs	0.1–1000 g/L	0.04 g/L	-	-	Tap water	2017, [60]
	rGo/AuNPs composite	5–100 μg/kg	2 μg/kg	9.5 min	73.4–92.8	Three kinds of thirty fried foods collected from 6 provinces in China	2019, [61]
	Strawberry-like SiO_2_/Ag nanocomposites (F-SANC)	0.1–50 μM	0.02 nM	-	80.5–105.6	Cookies, chips, and bread	2020, [13]
	AgNPs substrate	10–500 μg/L	2.5 μg/L	-	94.67–117.50	Potato chips	2023, [62]
	Core-shell structured Au@Ag NPs	10^−8^–10^−3^ mol/L	1.27 × 10^−9^ mol/L	-	85.68–102.50	Potato chips, fried dough twist, and instant coffee	2024, [68]

**Table 3 sensors-24-03501-t003:** Bio/chemical sensors based on electrochemical transduction.

Sensor Type	Type	Modifier_Electrode	Linear Range	LoD	Detection Time	Average Recovery (%)	Food Sample	Year, Refs
Voltammetric	Hb	Hb_Carbon-paste electrode	-	1.2 × 10^−10^ M	-	-	Potato chips	2007, [69]
		Hb/SWCNTs_GCE	1.0 × 10^−11^–1.0 × 10^−3^ M	1.0 × 10^−9^ M	-	-	Potato chips	2008, [70]
Coulometric	Hb	Hb/AuNPs_ITO glass	-	0.1 μM	-	-	-	2011, [71]
		c-MWCNT/CuNPS/PANI_PGE	5–75 mM	0.2 nM	2 s	95.40–97.56	Potato chips	2012, [72]
		MWCNTs/Fe_3_O_4_NPs/PANI_PGE	3–90 nM	0.02 nM	8 s	95.40–97.56	Potato chips	2013, [73]
Voltammetric	Cell	PC-12 cells/ERGO_GCE	0.1–5 mM	0.04 mM	-	-	-	2013, [74]
	DNA	DNA/GO_GCE	5.0 × 10^−8^–1.0 × 10^−3^ mol/L	-	-	-	-	2014, [75]
	MIT	P-ATP/AuNPs/PMA_MIP_GCE	0.5 × 10^−12^ mol/L	5 × 10^−13^ mol/L	-	above 95	Potatoes	2014, [76]
		MWCNTs/AuNPs/Ch_MIP_GCE	0.028 μg m/L	0.05–5 μg mL/L	-	84.7–94.8	Potato chips	2016, [77]
	DNA	ssDNA_AuE	8.1 nM	0.4–200 M	-	93.8–109.3	Tap water and potato chips	2016, [78]
	Hb	MWCNTS-IL/Ch-IL/PtAuPd NPs/Hb-DDAB_GCE	0.01 nM	0.03–150.0 nM	8 s	99.36–101.4	Potato chips	2018, [79]
		dsDNA(ssDNA1-Hb/ssDNA2-SPE)/Hb _SPGE	2.0 × 10^−6^–5.0 × 10^−2^ M	1.58 × 10^−7^ M	-	91–120	Potato fries	2019, [80]
	MIT	PPy/ZnO/AA(MIP)_FTO electrode	10^−1^–2.5× 10^−9^ M	2.147 × 10^−9^ M	-	92.64–106.0	Potato chips and cookies	2020, [81]
	Hb	Au@Ag CS-Hb/MXene/AuE	1–150 μM	3.46 μM	-	above 96	Sunflower oil	2022, [82]
	MIT	NOMG/3-TAA@AuNPs/PMA(MIP)_QCM chip	0.08–100 ng/mL	5.1 pg/mL	-	88.3–97.2	Bread, potato chips, and cookies	2022, [83]
	Nbs	XAA Nbs_SPCE	0.39–50.0 μg/mL	0.033 μg/mL	30 min	88.29–111.76	Piked baked biscuits and potato crisps	2022, [84]
	DNA	Adenine _BDD electrode	0.14–1.00 μM	0.10 μM	-	-	-	2023, [85]
	Hybrid	MIP-Apt-Au@rGO-MWCNTs_GCE	1–600 nM	0.104 nM	-	98.7–103.4	Potato fries	2023, [86]
	ECL	$Ru{\left(bpy\right)}_{3}^{2+}$@ZnO-Au(MIP)_GCE	1–$10^{8}$ nM	0.123 nM	-	93.3–104.7	Potato chips, cookies, and instant coffee	2024, [87]
		$Ru{\left(bpy\right)}_{3}^{2+}$_Pt electrode	5 μM–10 mM	1.2 μM	-	-	-	2019, [88]

#### 2.2.1. Hb Label-Based EC Sensor

It is known that AA and related conjugated vinyl compounds undergo Michael-type nucleophilic addition reactions of -$NH_{2}$ and -SH of amino acids, peptides, and proteins to their double bonds [89]. Investigations showed that the formation of AA-hemoglobin (AA-Hb) adducts through the reaction of AA with the α-NH_2_ group of the N-terminal valine in Hb [4,89]. Consequently, Hb can serve as a useful biomarker of human exposure to AA. However, there are two challenges: first, achieving the direct electron transfer between Hb and electrode proves challenging, and second, the direct adsorption of biomolecules to the bare surface of the bulk material may often lead to their denaturation and bioactivity loss.

The first challenge stems from the large, and three-dimensional structure of Hb, which renders the redox centers within the protein inaccessible. Therefore, the utilization of electromediators is crucial for accelerating the rate of electron transfer between Hb and the electrode surface. Stobiecka et al. pioneered a directly adsorbed biosensor comprising an Hb-modified carbon-paste electrode, leveraging the reaction between AA and Hb, achieving an LoD of 1.2 × 10^−10^ M in the presence of the crisp matrix. The formation of the Hb-AA adduct was observed by a reduction in the peak current of Hb-Fe(III), attributable to the diminished electroactivity of Hb due to the reversible conversion of Fe(III) to Fe(II) within the four hematin pseudogroups of Hb [69]. Yet, this directly adsorbed mode can easily lead to the Hb loss of biological activity.

For the second one, it can be solved by modifying electrodes with nanoparticles. Nanoparticles facilitate the adsorption of biomolecules while retaining their bioactivity. What is more, carbon nanotubes (CNTs) have emerged as a class material for immobilizing enzymes/proteins, attributed to their high electrical conductivity, superior chemical and mechanical stability, and large surface area [72].

Batra et al. reported a relatively stable biosensor for AA detection based on the synergistic effect of metallic NPs along with CNTs and conducting polymer [73]. In this study, a nanocomposite of carboxylated multiwalled carbon nanotubes (cMWCNT) and Fe_3_O_4_NPs was electrodeposited onto an Au electrode through a chitosan (Ch) film. Ch was chosen for its low toxicity, high biocompatibility, controllable film thickness, and high mechanical strength. Furthermore, the biosensor was constructed by immobilizing Hb on a cMWCNT/Fe_3_O_4_NPs/Ch-modified Au electrode. This biosensor demonstrated a relatively rapid response time (8 s), a broad linear range (3–90 nM), a low detection limit (0.02 nM), and relatively long-term stability [73]. However, its specificity has not been studied, raising the possibility of false positives or negatives in practical applications.

Ionic liquids (ILs) represent another class of materials for electrochemical applications due to their exceptional properties, including stability, high electrical conductivity, and low vapor pressure [90]. Varmira et al. proposed a multilayer composite films-based biosensor for AA determination in food samples. The biosensor platform utilized a GCE modified with three-layer composite films. The first layer consisted of MWCNTs incorporated with ILs (MWCNTs-ILs) to enhance electrical conductivity. Subsequently, platinum–gold–palladium (PtAuPd) alloy nanoparticles were electrodeposited onto a chitosan-ionic liquids (Ch-ILs) layer to absorb Hb. Finally, the biosensor was coated with a layer of Hb-dimethyldioctadecylammonium bromide (Hb-DDAB) to accelerate electron transfer and enhance the Hb concentration. In light of the matrix effect, the anti-interference performance of the proposed sensor was evaluated for common compounds found in food samples, including dinitrobenzaldehyde, acetaldehyde, hydrazine, phenol, acetic acid, ascorbic acid, tartrate, and bromobenzaldehyde. It was found that the presence of large amounts of these compounds had no significant interference on the AA determination, with the tolerance limit of interference being the concentration at which the error remains within 5.0%. Although the sensor exhibited high sensitivity (an LoD of 0.01 nM), a short response time (less than 8 s), good specificity, and reproducibility, its stability was affected by the duration of immersion in the cell solution [79].

In addition to the composite film for Hb immobilization described above, Asnaashari et al. presented an electrochemical method involving the conjugation of the amine-modified $ssDNA_{2}$ to the carboxyl groups of Hb by using the N-(3-dimethylaminopropyl)-n-ethyl-carbodiimide hydrochloride (EDC)/N-hydroxysuccinimide (NHS) technique. On the other hand, a screen-printed gold electrode ($ssDNA_{1}$-SPGE) was modified with $ssDNA_{1}$ containing a thiol group ($ssDNA_{1}$-SH). Therefore, Hb was immobilized on the SPGE through these two single-stranded DNA sequences, facilitating the recovery of the electrode. Although the method possessed an LoD of 1.58 × 10^−7^ M and a wide dynamic range, the presence of the potato fries matrix faintly impacted its sensitivity to AA [80].

Recently, Divya et al. used MXene nanosheets to form a stable substrate-modified Au electrode. The electrode was then fixed with the Au@Ag core-shell and Hb complex. The sensor possessed a simple structure, but its performance was not as good as that of the aforementioned research [82].

#### 2.2.2. Immunosensors

Immunoassays, relying on specific interactions between an antibody and corresponding antigen, can meet the analytical requirement of a useful AA assay with high sensitivity, cost-effectiveness, and specificity. However, due to AA’s low molecular weight and the lack of strong epitope groups, antibody production against AA and the establishment of related immunoassays are challenging [91]. To avoid the loss of limited epitopes that may result from the direct conjugation of AA to carrier proteins, Zhou et al. synthesized polyclonal antibodies (pAbs) with NAS as haptens rather than AA itself. A biotin-avidin enzyme-linked immunosorbent assay (BA-ELISA) was then established based on the pAbs. However, the assay exhibited a low degree of cross-reactivity (CR) with methacrylamide, methyl acrylate, and acrylonitrile [92].

Furthermore, the repeatability of the above assays has been very poor, as attempts to produce specific AA antibodies using the same strategy have failed twice in Wu’s group. Therefore, Wu et al. produced a pAb targeting another derivative of AA, 4-mercaptophenylacetic acid-derivatized AA (AA-4-MPA), and developed a competitive indirect ELISA (ci-ELISA) for AA via preanalysis derivatization [91]. Cross-reactivity tests against a series of structural analogs and their derivatives, including 4-MPA, demonstrated that the antibody has high specificity against AA-4-MPA. Building on this, Wu et al. combined the AA-4-MPA specific polyclonal antibody conjugated on gold nanorods (AuNR) as a primary antibody (AuNR-Ab1) and horseradish peroxidase-labeled antirabbit antibody conjugated on AuNR as a secondary antibody (HRP-AuNR-Ab2), thereby significantly improving the electrochemical signal. SnO^2^-SiC hollow spherical nanochains with good catalytic activity and durability were then modified onto a glassy carbon electrode (GCE) using chitosan to further amplify the signal. The proposed immunosensor exhibited good selectivity with no cross-reaction, achieving an LoD of 45.9 ± 2.7 ng/kg. Not only can the electrode not be reused, but the derivatization reaction between AA and MBA involved in the fabrication requires hyperthermia (>50 °C) and a long wait time (>1 h) [93].

To make the immunoassay simpler and faster, Xanthydrol, which converts AA to XAA at room temperature within 30 min, can be employed. Concurrently, Luo et al. utilized 9-xanthydrol as a derivative to synthesize two haptens, named XAA-295 and XAA-309, against xanthydrol-derivatized AA (XAA) for the first time. Subsequently, they obtained a specific and high-affinity pAb against XAA (anti-XAA pAb) and developed the APL-based ELISA platform mentioned in Section 2.1.2 by using anti-XAA pAb as the recognition reagent [45].

Nanobodies (Nbs), derived from the heavy chain-only antibody in Camelidae [94], offer distinct advantages over traditional pAbs or monoclonal antibodies (mAbs) in terms of size, stability, production efficiency, engineering flexibility, reduced immunogenicity, and functionality in diverse environments [95,96]. Recently, Liang et al. successfully isolated a specific Nbs against XAA from a camel-immunized nanobody library for the first time. Following four rounds of panning and sequence alignment analysis, Nb-7E, exhibiting the highest inhibition rate, was chosen for further analysis. In this study, using anti-XAA mAbs as a control, it was found that anti-XAA Nb-7E showed better thermal stability and tolerance to methanol but poor tolerance to acetonitrile. Although Nb-7E does not recognize 9-xanthydrol, it can recognize the analog methyl carbamate, which may result in false positives when applied to practical applications. Subsequently, a ci-ELISA for AA was established based on Nb-7E. Furthermore, they constructed an enhanced electrochemical immunoassay (ECIA) biosensor with a wider linear range (0.39–50.0 μg/mL), lower LoD (0.033 μg/mL), and improved sensitivity. Finally, the analytical performance of the ECIA was validated by standard UPLC-MS/MS, suggesting that the proposed Nbs could be used as a novel reagent in immunoassays. The methods were effective and promising for AA detection in foodstuffs [84].

#### 2.2.3. MIP-Based EC Sensors

While nanomaterials, their complexes, and polymers can enhance immobilization efficiency and elevate the sensitivity, specificity, and detection capabilities of AA, their structure may be susceptible to extensive damage when exposed to high concentrations of AA. For instance, Gonzalez et al. developed a sensor using screen-printing electrodes (SPEs) modified with cSWCNTs, onto which AA got adsorbed. However, the hexagonal structure of CNTs was destructed in the presence of high AA concentrations, which adversely affected the detection process [97].

The molecular imprinting technique (MIT), a technique for generating template-shaped cavities (recognition sites) in polymer matrices, has been widely used in various fields because of its high selectivity, robustness, stability, reusability, cost-effectiveness, and customization. In this case, AA and its structural analogs act as templates [98]. The sensitivity of MIT sensors is determined by the number of effective recognition sites in the molecularly imprinted polymer (MIP) films and their conductivity [99,100]. Furthermore, the sol-gel imprinting method improved the performance of MIP film on sensor surfaces by facilitating control over the thickness, porosity, and surface area of the film [101]. Liu et al. developed an MIP sensor by combining a AuNPs-MWCNTs-Ch composite with sol-gel MIT for convenient and sensitive AA detection, achieving an LoD of 0.028 μg mL^−1^ [77]. However, creating more binding sites merely by increasing the imprinting film thickness results in slow diffusion of the analyte to recognition sites, leading to inefficient communication between imprinted sites and transducers [102,103]. Wang et al. made use of conductive polymers and doped them with metal nanoparticles to improve the conductivity of MIP sensors. The proposed surface MIP-based sensor employed p-thiophenol (P-ATP) as a functional monomer, propanamide (PMA) as an imitation template, and AuNPs as a crosslinker. PMA was chosen as the template molecule because it is challenging to elute AM from MIP, which can lead to high false positives [76].

Conductive polypyrrole (PPy) can be employed to enhance the specificity of photoelectrochemical (PEC) sensors due to its straightforward production good controllability, photoelectrochemical activity, and capability to form MIP films with inorganic semiconductors. PEC sensors utilize specific light to excite photoactive materials, using the electrical signal as the detection signal, which effectively separates the excitation and detection signals. These sensors feature simple equipment, a fast response speed, and ease of operation [104,105]. Zhao et al. reported an MIP-based PEC sensor using ZnO nanodisks as the photoactive material, PPy as the functional polymer, and AA as the template. Upon the presence of AA, the ZnO/PPy surface hindered the electron transfer, reducing the photocurrent signal as the detection signal. The proposed sensor exhibited a wide detection range, low detection limit (2.147 × 10^−9^ M), anti-interference ability, relatively easy setup, and a satisfactory recovery rate when applied to potato chip and cookie detection [81]. However, whether the designed electrode can be reused and the potential issues with AA exposure still need attention.

The QCM chip, featuring label-free detection and real-time digital output [106], is integrated into the electrochemical sensor. This advancement enables the sensing platform to be more effectively developed for the real-time and on-site monitoring of food safety risks [107,108]. Chi et al. established a highly sensitive EC detection platform with an LoD of 5.1 pg/mL by integrating QCM chips with MIT. This study used AuNPs (NOMC-Au) modified with nitrogen-doped OMC (NOMC) as the functional modification layer of QCM gold chips to provide more recognition sites. Additionally, the 3-thiopheneacetic acid (3-TAA) functionalized AuNPs (3-TAA@AuNPs) and PMA as a cross-linked imprinted layer formed an eluted three-dimensional network structure with a specific recognition ability toward AA. Using the functional QCM chip as the working electrode, the mass change caused by specific adsorption of AA on the MIP layer was converted into a digital frequency signal output for quantitative analysis. When the functionalized electrode was kept at 4 °C for 15 days, the frequency response decreased by 9.7%, indicating that the stability was not sufficient. The sensor demonstrated good reproducibility, repeatability, and selectivity, and could directly output digital signals, suggesting its potential for real-time AA detection in food samples [83].

#### 2.2.4. Electrochemiluminescence (ECL)-Based Sensors

Electrochemiluminescence (ECL) refers to a specific chemiluminescence reaction initiated by electrochemical methods on the electrode surface. This process involves the production of electrogenerated substances on the electrode surface upon the application of voltage. These substances undergo interactions and produce the luminescent body’s excited state. Upon returning to the ground state, the excited state emits light at a specific wavelength [109,110,111]. Yang et al. pioneered the utilization of ECL for AA detection. Under optimal conditions, Pt electrodes were employed with a scanning potential range of 0.02–0.1 V/s. During the process, AA, acting as a reducing agent, was oxidized when the scanning potential approached 1.1 V. Simultaneously, $Ru{\left(bpy\right)}_{3}^{2+}$, serving as the luminophore, was also oxidized to $Ru{\left(bpy\right)}_{3}^{3+}$, which reacted with oxidized AA to obtain the excited state $Ru{\left(bpy\right)}_{3}^{2+*}$. Upon returning to the ground state $Ru{\left(bpy\right)}_{3}^{2+}$, the unstable $Ru{\left(bpy\right)}_{3}^{2+*}$ emitted a significant ECL signal, appearing as orange–red visible light. Using a remote wireless camera, the red, green, and blue (RGB) model was converted to the hue, saturation, and value (HSV) model using MATLAB software, https://ww2.mathworks.cn/en/products/matlab.html (accessed on 26 May 2024). The proposed sensor demonstrated a wide detection range (5 μM–10 mM), an LoD of 1.2 μM, and satisfactory stability and reproducibility. Nevertheless, its specificity is not clear, though it provides a new approach for AA detection in food safety and other fields [88].

The combination of MIP with the ECL method offers enhanced selectivity and sensitivity. Kuang et al. recently reported a novel MIP-ECL sensor based on $Ru{\left(bpy\right)}_{3}^{2+}$@ZnO-Au composite material for AA detection. In this MIP-ECL immunosensor, $Ru{\left(bpy\right)}_{3}^{2+}$, ZnO-Au, and AA functioned as the luminescent material, substrate, and template molecule, respectively (Figure 6A). Upon excitation, the $Ru{\left(bpy\right)}_{3}^{2+*}$@ZnO-Au emits orange–red light around 620 nm upon returning to its $Ru{\left(bpy\right)}_{3}^{2+}$@ZnO-Au state in the presence of AA, triggering the ECL reaction. Additionally, ZnO-Au, acting as a carrier material, enhanced the ECL response by increasing the load of $Ru{\left(bpy\right)}_{3}^{2+}$. In addition to its wide linear range (1–$10^{8}$ nM), the LoD of 0.123 nM, excellent stability, repeatability, and high selectivity, the MIP-ECL sensor showed good enough recoveries when applied to potato chips and cookies, making it suitable for AA detection in real food samples [87].

#### 2.2.5. Label-Free DNA-Based EC Sensors

The density functional theory (DFT) analysis shows that AA acts not only as a good hydrogen bond acceptor but also as a hydrogen bond donor to high electronegative atoms such as oxygen (O) and nitrogen (N). Consequently, AA can establish stable hydrogen bonds with purine and pyrimidine bases of DNA, resulting in the formation of a stable DNA-AA adduct [112]. The bases within the double helix structure of DNA have been revealed to be electrochemically active, enabling direct electrochemical signal generation for DNA-based electrochemical biosensors without the need for additional electrochemical labels [112]. Building upon this understanding, Li et al. proposed a label-free DNA biosensor for electrochemical AA detection by immobilizing dsDNA on a GCE modified with graphene oxide (GO). Since the peak current of guanine is higher than that of adenine, guanine was chosen as a monitoring signal in subsequent studies. The electrochemical signal of DNA bases immobilized on the electrode surface declined in the presence of AA, achieving a linear range (5.0 × 10^−8^–1.0 × 10^−3^ mol/L) [75].

Since AA primarily forms single and stable adducts with the guanine base of dsDNA at the N-7 position [113,114,115], these DNA-based biosensors exhibited high sensitivity and strong specificity. Even so, the increased steric hindrance of dsDNA inhibits the electron transfer efficiency. Therefore, Huang et al. proposed for the first time a convenient ssDNA-based biosensor for direct AA determination, utilizing ssDNA with a good base sequence to enhance the analytical performance of the sensor. Upon the presence of AA, the strong bond formed between AA and the guanine base of ssDNA at the N-7 position led to the formation of AA-ssDNA adducts, inhibiting the electroactivity of ssDNA. Consequently, the decrease in the differential pulse voltammogram (DPV) peak current was utilized for the selective and sensitive determination of AA. This ssDNA/GE biosensor exhibited a simple and fast response to AA, with long-term stability and excellent reproducibility. Under optimal conditions, the detection limit of AA was 8.1 nM [78]. Recently, Anggraini et al. utilized a combination of molecular docking simulations and wet experimental methods to identify free purine bases as new biomarkers for AA detection. In the coffee sample, they achieved an LoD of 0.01 μM using adenine as a case study [85].

#### 2.2.6. Label-Free Cell-Based EC Sensors

Cell-based biosensors have been used in label-free and real-time monitoring technologies due to their simplicity, sensitivity, and cost-effectiveness [116]. Amidase (acylamide amidohydrolase; E.C.3.5.1.4) derived from Pseudomonas aeruginosa exhibits transferase and hydrolase activities with aliphatic amide substrates, including AA [117]. This enzyme catalyzes the transfer of acyl groups from amides to hydroxylamine, forming acyl hydroxamates and ammonia as well as the hydrolysis of aliphatic amides such as AA, resulting in the corresponding acid and ammonia [118]. Thus, Silva et al. reported a low-cost biosensor based on an ammonium ion-selective electrode (ISE) and immobilized whole cells of Pseudomonas aeruginosa containing amidase activity for AA detection. However, employing a single polymer film as support for whole-cell immobilization led to considerable biomass loss, resulting in a premature reduction in the biosensor’s overall activity [119]. To address this issue, Silva et al. then adopted a “sandwich” design employing two membrane discs to prevent premature loss [120]. Even so, the challenge of using polymeric membranes in the immobilization process persisted. Surface functionalization using biocompatible materials has emerged as a promising approach for cellular adhesion without compromising viability. Sun et al. devised a label-free cell-based electrochemical sensor by immobilizing living pheochromocytoma (PC-12) cells on an AuNPs/electrochemically reduced graphene oxide (ERGO) self-assembled GCE. This sensor enabled the monitoring of AA in vitro within a range of 0.1–5 mM, with an LoD of 0.04 mM [74]. In addition to the unclear specificity of the aforementioned sensors, their most obvious drawback is their inability to provide a quantitative measurement of AA in food samples. This is mainly due to their limited lifespan and potential loss of stability even after initial exposure to minimal concentrations of AA.

## 3. Recent Development

### 3.1. Nanozyme-Based Colorimetric Sensor

Enzyme-based colorimetric detection methods exhibited excellent sensitivity and strong operability owing to their inherent exponential amplification manner of signals. Unlike natural enzymes, nanozymes have attracted significant research interest due to their intrinsic enzyme-like activity, especially Fe_3_O_4_ nanoparticles (FeNPs) with the peroxidase (POD)-like activity of FeNPs [121]. Liu et al. proposed a hedgehog-like Fe^3+^-polydopamine hierarchical superstructures (Fe-PHSs) sensor for the colorimetric determination of AA. This innovative approach relied on the metal–ligand cross-linking strategy and the thiol-ene Michael addition reaction, achieving an LoD of 0.27 μM. Remarkably, this study marked the first application of Fe-PHSs in developing colorimetric sensors to mimic enzymatic activity. Leveraging its ultrahigh POD-like activity, the hedgehog-like Fe-PHSs effectively catalyzed and oxidized colorless 3,3^′^,5,5^′^-tetrame-thylbenzidine (TMB) to ox-TMB in the presence of H_2_O_2_, with an absorption peak at 652 nm. The oxidation of TMB was impeded by GSH but can be restored through a thiol-ene Michael addition reaction between GSH and AA. Consequently, higher concentrations of AA result in accelerated TMB oxidation, accompanied by a darker blue color recovery. In the range of 0.75–36 μM, the UV absorbance at 652 nm increased linearly with the increase in the AA concentration, and the LOD met the AA detection requirements in food processing (31–739 ppb). Although the sensor possessed relatively prolonged storage stability, its storage conditions (4 °C) are strict. The Fe-PHS sensor was not only highly selective, fast, low-cost, and suitable for visual detection, but the chemicals required for its fabrication were readily available, making it convenient for on-site detection [66].

However, systems relying on individual enzymes often suffer from cumbersome operational steps, inefficiency, and protracted processes. The MOFs represent a class of inorganic–organic hybrid microporous crystal materials with three-dimensional structures composed of metal ions and organic ligands, renowned for their high porosity and exceptional stability. To address these limitations, natural enzymes can be incorporated into the MOFs to construct the enzyme cascade system. Additionally, aptamers (Apts) have been implemented to diversify the range of analytes. Thus, Guo et al. devised a colorimetric aptasensor based on Apts and the MOF-enzyme cascade for AA detection, achieving an LoD of 1.53 nM. They established the enzyme cascade system MIL-GOx (MGzyme) by embedding glucose oxidase (GOx) in MIL-101. As depicted in Figure 7A, the presence of AA caused the dispersion of MGzyme-csDNA in the supernatant after magnetic separation. Upon adding glucose and TMB to the supernatant, the redox reaction of glucose and GOx in exfoliative MGzyme-csDNA produced H_2_O_2_ in situ and gluconic acid as a byproduct. MIL-101 oxidized H_2_O_2_, generating a radical (•OH) that oxidized colorless TMB to blue ox-TMB. The specific capture of Apts, along with the suitable microenvironment and reduced steric hindrance created by the enzyme cascade system, enhanced the sensitivity of the colorimetric aptasensor. After preparing the samples for testing, this aptasensor could realize the detection within 50 min [65]. As illustrated in Figure 7B–D, the colorimetric aptasensor demonstrated a wide range of enzyme cascade systems, high anti-interference, and selectivity, indicating significant potential for food safety inspections.

### 3.2. UCNPs-Based Aptasensor

Upconversion nanoparticles (UCNPs), a type of luminescent nanomaterial, consist of inorganic crystals doped with rare-earth ions [122]. They have the unique ability to convert lower-energy, longer-wavelength excitation light, such as near-infrared light, into higher-energy, shorter-wavelength emission light, such as visible or ultraviolet light [123]. UCNPs have gained significant attention for detecting potentially harmful compounds or contaminants in food items, due to their high photostability, minimal autofluorescence interference, and biocompatibility to overcome limitations associated with traditional fluorescent probes [124,125]. FRET, a universal tactic for UCNPs-based sensors, has been extensively utilized. Rong et al. pioneered the use of upconversion-based aptamer sensors for AA detection in food, with an LoD of 1.13 nM [126].

To advance on-site detection, there has been a surge in research interest in solid-phase detection systems [127]. Polydimethylsiloxane (PDMS), commonly used in biodevice fabrication due to its elasticity, can accommodate fluorescent materials without affecting their optical characteristics [128]. Integrating PDMS with fluorescent analysis offers a feasible and visually intuitive method for POC analysis. Hence, Rong et al. designed a maneuverable solid-state fluorescent sensor suitable for both spectral and visual monitoring, showcasing an LoD of 1.00 nM and 1.07 nM, respectively. As depicted in Figure 8A, PDMS served as the substrate uniformly coated with core-shell UCNPs (CSUCNPs) as the reporter, modified with csDNA of Apts (CSUCNPs@PDMS). The nanocrystalline fluorescein isothiocyanate isomer (FITC) was conjugated with Apts, acting as a quencher (Apt-FITC@SiO_2_). The binding of FITC@SiO_2_ to CSUCNPs induced quenched upconversion luminescence via FRET. Upon the presence of AA, the detachment of FITC@SiO_2_ and rapid restoration of luminescence ensued due to van der Waals forces and hydrogen bonds between the Apts and the target AA [67]. Through the analysis of Figure 8B–F, they proved that the sensor exhibited good performance and held promise for utilization in portable devices for the on-site detection of hazardous substances in food processing.

### 3.3. Hybrid Multirecognition-Controlled Sensors

Ali et al. first proposed a novel dual-recognition electrochemical aptasensor for detecting AA in potato fries, utilizing a combination of Apt and MIP. As shown in Figure 9A, the aptasensor was constructed by depositing Au/rGO-MWCNTs onto the surface of a GCE. Then, the modified electrode was incubated with aptasensor and AA, followed by the electropolymerization of the o-phenylendiamine monomer to form MIP/Apt-SH/ Au@rGO-CNTs/GCE. This dual-recognition-controlled biosensor demonstrated high selectivity, reliability, a low detection limit (0.104 nM), and reasonable stability (Figure 9B,C). By specifically sensing AA in different matrices without interference from coexisting species, the sensor holds promise for on-site detection in food applications [86].

## 4. Conclusions and Outlook

The adoption of AA mitigation strategies in food processing, mandated by regulatory agencies such as the U.S. FDA and the EU Commission, has sparked concerns among both countries and consumers regarding public health and food safety. Consequently, numerous biological/chemical sensors with high sensitivity, selectivity, stability, reproducibility, low cost, minimal response time, and sustainability are emerging as indispensable tools for the convenient real-time and on-site assessment of AA levels in food processing. By analyzing the advantages and limitations of these sensors for AA detection over the past decade in terms of the sensing mechanism, linear range, detection limit, selectivity, and robustness, the potential development directions for these sensors are as follows:Improving sensitivity: Employing more amiable receptors/ligands and nanomaterials with high conductivity, biocompatibility, and large surface areas such as nanozyme, ZnO, CNTs, MXene, and alloy nanoparticles significantly enhance the interaction between AA and the probe surface. Alternatively, various signal-amplification techniques, such as enzyme labels, ECL, SPR, microfluidic systems, and ratiometric fluorescence strategies, and even multiplexing technology, could be utilized.Strengthening anti-interference: Use highly selective recognition elements, such as antibodies, aptamers, or MIPs, that bind specifically to AA while discriminating against other substances. Blocking agents, such as BSA, could be applied to prevent non-specific binding. Implement advanced sample preparation techniques such as SPE, dSPE, and QuEChERS and calibration techniques such as matrix-matched calibration and the standard addition method to eliminate matrix effects.Enhancing reproducibility: Using reliable coating technologies, such as graphene-based substrates based on the GERS effect, to achieve a uniform response. Employing composite materials could integrate the advantages of various components to enhance stability and reproducibility.Advancing sustainability: Utilizing biodegradable or recyclable materials for sensor components and choosing eco-friendly, non-toxic, or less toxic alternatives to avoid hazardous substances. Implementing waste reduction strategies in the production process, such as recycling byproducts. Establishing modular components that are easy to replace or upgrade could extend the lifespan of the sensor.Easy to use: Establishing direct sampling or integrated sample handling steps to minimize sample preparation. Integrating portable bio/chemical sensing arrays with smartphone sensing techniques and other mobile devices.

By leveraging these advancements, bio/chemical sensors will increasingly become used to ensure food safety and address growing concerns related to AA contamination in food products.

## Figures and Tables

**Figure 1 sensors-24-03501-f001:**
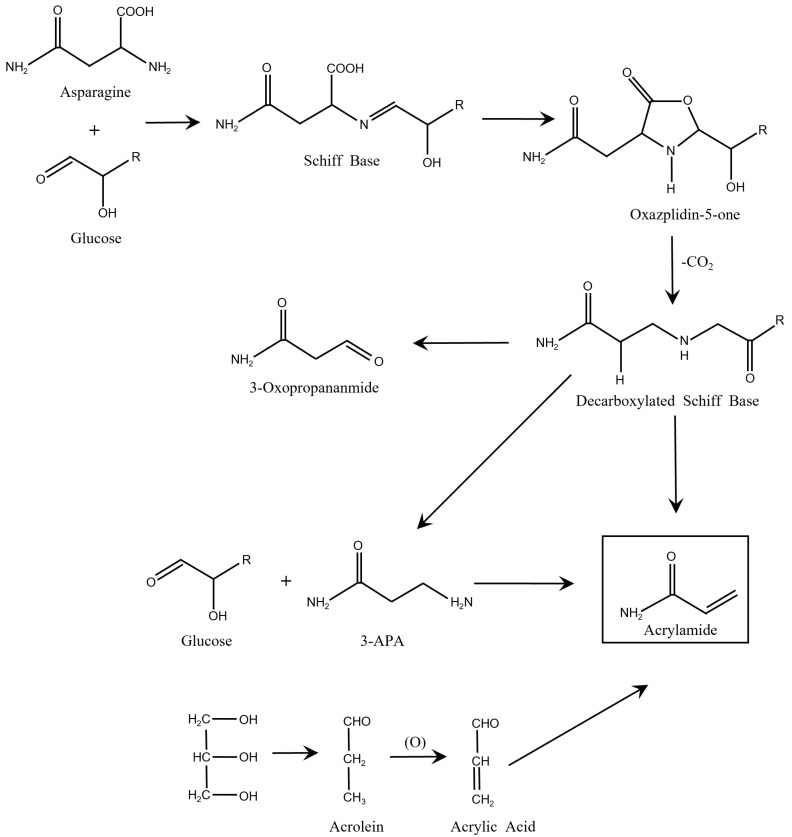
Illustration of formation mechanisms of AA (Maillard reaction and acrolein way) [5]. Copyright 2015 Elsevier.

**Figure 2 sensors-24-03501-f002:**
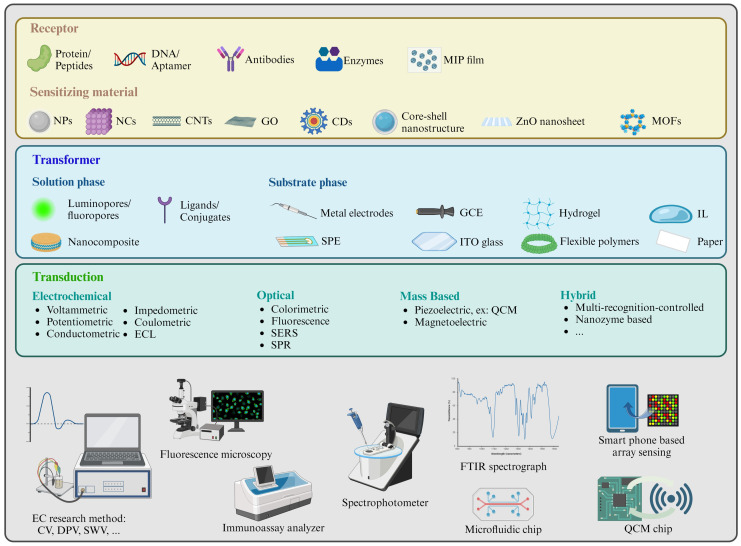
Illustration of overview of bio/chemical sensing tech in AA detection.

**Figure 3 sensors-24-03501-f003:**
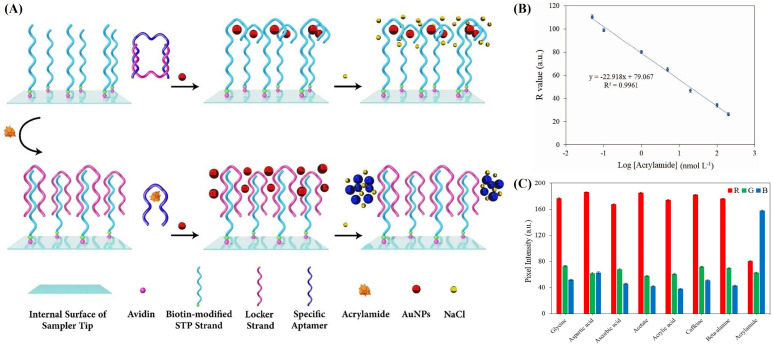
(**A**) Illustration of the colorimetric aptasensor embedded in a micropipette tip. (**B**) The calibration plot of the R parameter and AA concentration (0.05–200 nmol L^−1^). (**C**) Quantitative RGB analysis of the solutions in the tips. Permission from [35]. Copyright 2023 Elsevier.

**Figure 4 sensors-24-03501-f004:**
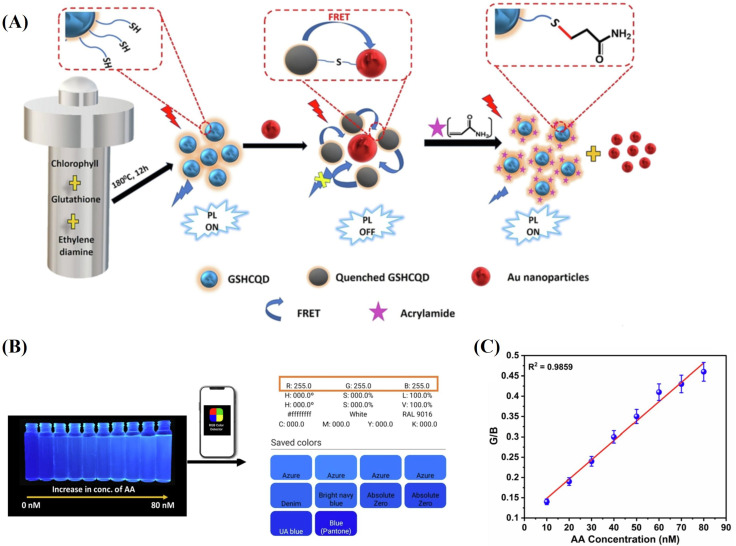
(**A**) Illustration of synthesis of GSHCQD–Au nanoprobe and its AA detection. (**B**) RGB analysis of GSHCQD–Au probe solution under prolonged UV exposure after AA addition. (**C**) The plot of linearity between the G/B value and the AA concentration. Permission from [42]. Copyright 2023 Elsevier.

**Figure 5 sensors-24-03501-f005:**
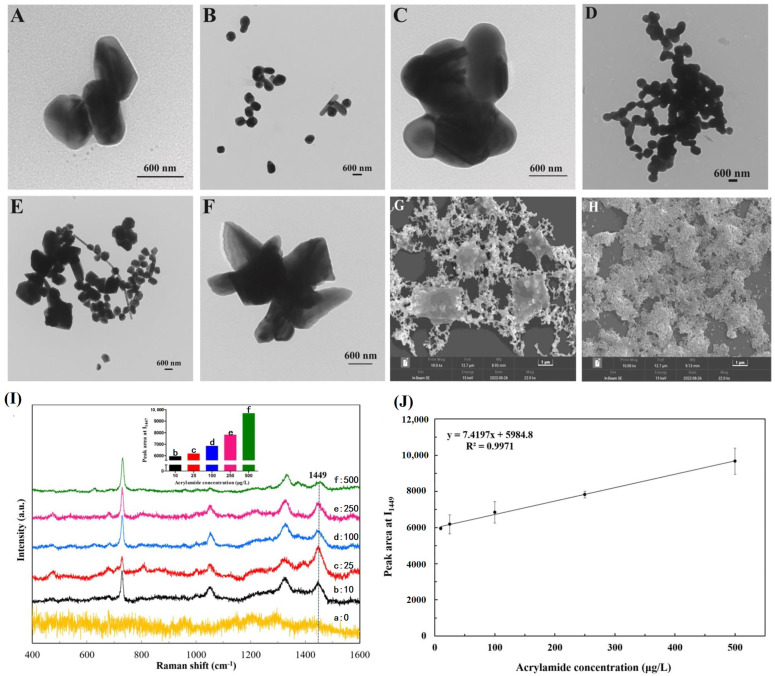
TEM and SEM characterization of AgNPs (**A**), AgNPs + KCl (0.05 M) (**B**), AgNPs + MgCl_2_ (0.05 M) (**C**), AgNPs + NaCl (0.5 M) (**D**), AgNPs + MgSO_4_ (0.05 M) (**E**), and AgNPs + Na_2_SO_4_ (0.05 M) (**F**). SEM images of AgNPs (**G**) and AgNPs + NaCl (0.5 M) (**H**). (**I**) SERS spectra of various concentrations of AA standard solutions. (**J**) Calibration curve of peak area at 1449 cm^−1^. Permission from [62]. Copyright 2023 Elsevier.

**Figure 6 sensors-24-03501-f006:**
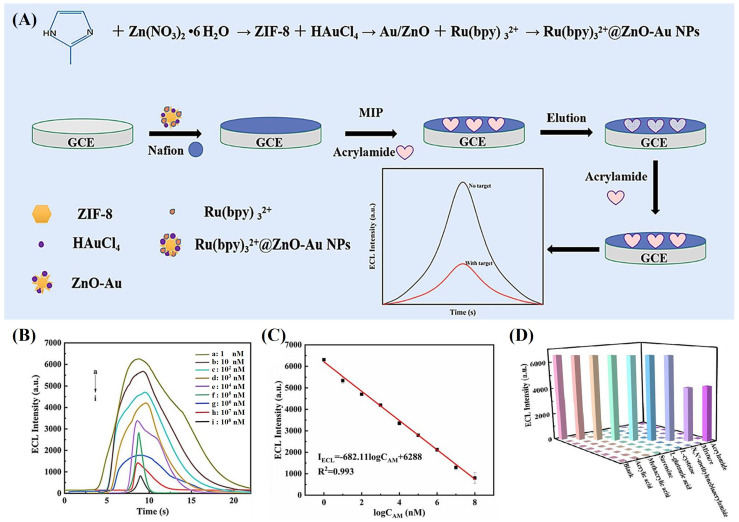
(**A**) The fabrication of molecular–imprinted ECL sensor based on $Ru{\left(bpy\right)}_{3}^{2+}$@ZnO–Au. (**B**) Quantitative AA detection by MIP–ECL sensor. (**C**) Linear calibration curve of different concentrations of AA. (**D**) The specificity of MIP–ECL sensor. Permission from [87]. Copyright 2024 Elsevier.

**Figure 7 sensors-24-03501-f007:**
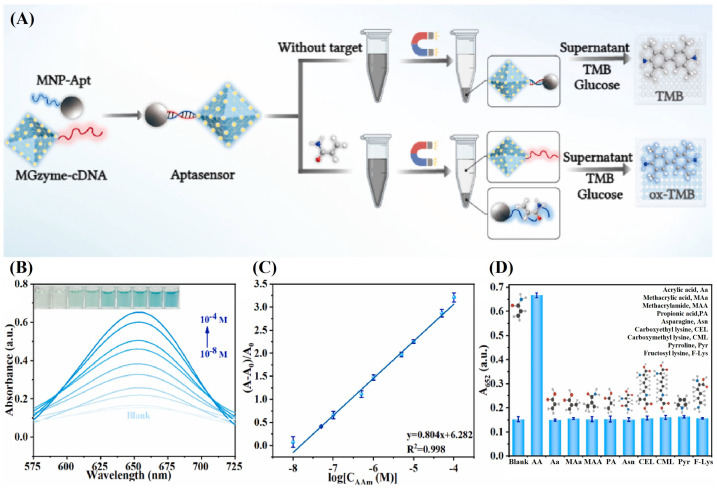
(**A**) UV–vis Absorption spectra of the detection system upon the addition of different concentrations of AA. (**B**) The standard curve of relative absorbance at 652 nm versus logarithmic concentrations of AA. (**C**) Investigations of specificity and anti-interference capability of colorimetric aptasensor (**D**). Modified and adapted with permission from [65]. Copyright 2024 Elsevier.

**Figure 8 sensors-24-03501-f008:**
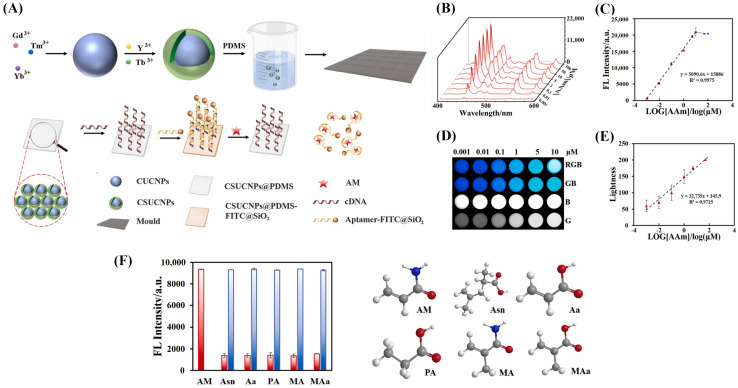
(**A**) Illustration of the UCNPs–based sensor for AA. (**B**) The upconversion emission spectrum in different AA concentrations. (**C**) Linear association between the fluorescence intensity at 450 nm and the logarithm of AA concentrations. (**D**) Luminescence images of the UCNPs–based sensor. (**E**) Linear association between the brightness of images and the logarithm of AA concentrations. (**F**) Selectivity and the anti-interference assay. Permission from [67]. Copyright 2024 Elsevier.

**Figure 9 sensors-24-03501-f009:**
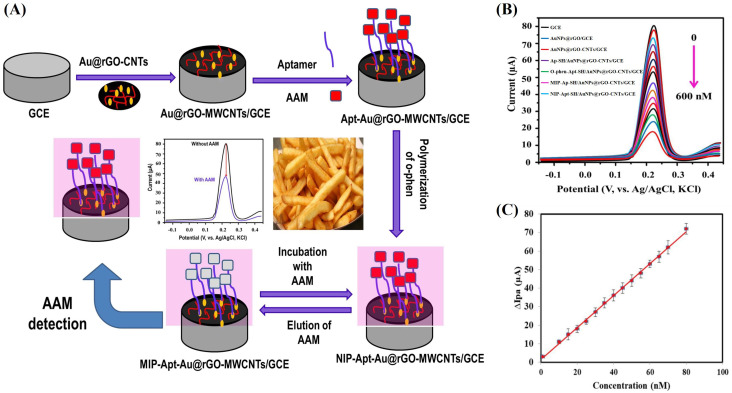
(**A**) Main steps in preparation of the aptasensor and detection of AA. (**B**) The DPV scans of the aptasensor for different amounts of AA. (**C**) DPV scans were performed in 0.1 M PB containing 2.5 mM [Fe(CN)_6_]^3−/4−^ under optimum conditions. Modified and adapted with permission from [86]. Copyright 2023 Elsevier.

**Table 1 sensors-24-03501-t001:** Typically employed receptors and ligands in sensor fabrication.

Ligand	Receptor	Adduct and Complex
Acrylamide	Cysteine	Cysteine-acrylamide adduct
	Glutathione	Glutathione-acrylamide adduct
	Thiol group functionalized oligonucleotide	Thiol group functionalized oligonucleotide-acrylamide adduct
	Guanine	Guanine-Acrylamide adduct
	Fluorescein	Fluorescein-Acrylamide derivative
	Fluorescamine	Fluorescamine-Acrylamide derivative
	Acrylic acid	Acrylic acid-Acrylamide complex via hydrogen bonding
	Carboxylic acid radical	Carboxylic acid radical mediated acrylamide complex
	Hemoglobin (in Fe^2+^ state)	Hemoglobin valine mono- and bis-adducts

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
