# Peer review of "Advancements in Chemical and Biosensors for Point-of-Care Detection of Acrylamide"

_sensors, 2024, doi:10.3390/s24113501_

Round 1

Reviewer 1 Report

Comments and Suggestions for Authors

This review paper provides a comprehensive overview of recent advancements in bio/chemical sensors for the detection of acrylamide in food products, focusing on sensing mechanisms, selectivity, linear range, detection limits, and robustness. The authors discuss various types of sensors, including optical, electrochemical, hybrid, and piezoelectric transduction systems, and evaluate their strengths and limitations.

1. Further expending the key characteristics of the discussed sensors for quick comparison. Include a column for the food matrix used in each study.

2. Discuss the matrix effects and potential interferences in real food samples for each type of sensor. Provide insights into how these challenges can be addressed.

3. Compare the performance of the bio/chemical sensors with conventional analytical methods (e.g., LC-MS/MS, GC-MS) in terms of sensitivity, selectivity, and analysis time. Highlight the advantages and limitations of each approach.

4. Critically evaluate the scalability and cost-effectiveness of the sensor fabrication processes. Identify potential bottlenecks in their commercial application.

5. Reduce the similarity. Its almost 50%.

Author Response

Dear Reviewer,

We would like to thank you for your evaluation of our paper and for providing us with many helpful comments and suggestions. We have seriously considered these comments and suggestions, and carefully revised the manuscript accordingly. The details are explained in the attachment (response numbers are in 1:1 correspondence with the comments). For the convenience of the Reviewers’ checking, all the corrections and modifications are marked in color in the revised version.

Again, we appreciate the Reviewers’ encouraging evaluations of our work.

Sincerely yours,

Mingna Xie, Xiao Lv, Ke Wang, Yong Zhou, Xiaogang Lin.

Reviewer 2 Report

Comments and Suggestions for Authors

The article provides a comprehensive examination of various bio/chemical sensors developed over the past decade for detecting acrylamide (AA) in thermally processed foods. AA is noted for its potential carcinogenic, neurotoxic, and reproductive hazards. The paper excels in detailing the advancements in sensor technology, focusing on their reliability, sensitivity, selectivity, convenience, and cost-effectiveness. It methodically categorizes sensors based on their sensing mechanisms, such as optical, electrochemical, and others, further subdivided into specific types like colorimetric, fluorescence, SERS sensing, and more. The review concludes with a discussion on the potential developments for point-of-care applications, underscoring the importance of these technologies in enhancing food safety.

The paper is well-structured and informative, providing valuable insights into the state of AA detection technology. However, to strengthen the manuscript and make it an essential reference in the field, the authors should consider addressing the following comments and questions. 

1. How do the discussed sensors specifically distinguish AA from other similar compounds present in food samples? Detailing the specificity of these sensors will help in understanding their practical applications and limitations regarding false positives or negatives.

2. It would be advantageous to include data or case studies that showcase the performance of these sensors under real-world operational conditions within food processing industries. 

3. Address the environmental implications related to the disposal and lifecycle management of these sensors, especially those utilizing hazardous or bioactive materials.

Author Response

(The authors gave the same response as above.)

Round 2

Reviewer 1 Report

Comments and Suggestions for Authors

The revised version of the manuscript fulfills all the criteria for publication. I endorse its acceptance.